# Unequal Access to Newly Registered Cancer Drugs Leads to Potential Loss of Life-Years in Europe

**DOI:** 10.3390/cancers12082313

**Published:** 2020-08-17

**Authors:** Carin A. Uyl-de Groot, Renaud Heine, Marieke Krol, Jaap Verweij

**Affiliations:** 1Erasmus School of Health Policy & Management, Erasmus University Rotterdam, Burg Oudlaan 50, 3062 PA Rotterdam, The Netherlands; heine@eshpm.eur.nl; 2IQVIA, Herikerbergweg 314, 1101 CT Amsterdam, The Netherlands; Marieke.Krol@iqvia.com; 3Department of Medical Oncology, Erasmus Medical Center, Dr. Molewaterplein 40, 3015 GD Rotterdam, The Netherlands; jaap@cddf.org

**Keywords:** cancer drugs, access, delay, inequality, life years

## Abstract

Background. Many new cancer medicines have been developed that can improve patients’ outcomes. However, access to these agents comes later in Europe than in the United States (US). The aim of this study is to assess the access in Europe to newly registered cancer drugs and to get more insight in the implications of these variations for patients. Methods. A retrospective database study was conducted. Analyses involved 12 cancer drugs and 28 European countries in the period 2011–2018. Time to patient access, speed of drug uptake, and the potential loss of life years due to a delay in access have been studied. Results. Marketing approval for the cancer drugs came on average 242 days later in Europe than in the US, and actual patient access varied extensively across Europe. The average time to market in Europe was 403 days (range 17–1187 days). The delay in patient access of ipilimumab and abiraterone may have led to a potential loss of more than 30,000 life years. Conclusion. It takes a long time for patients to get access to newly registered cancer drugs and there is great variation in access. The health outcomes can be substantially improved by faster processes.

## 1. Introduction

Cancer is a major cause of death and therefore a pressing international public health concern [1,2]. Cancer incidence is increasing in all European countries (EC). Sales of cancer drugs have more than doubled between 2005 and 2014 [3]. Because of the recent scientific advances, many new drugs have been developed that can improve overall survival (OS), prolong time to tumor progression (TTP), or decrease the chance of recurrence of cancer [4]. However, access to those drugs is not equal across Europe, as the time from a marketing approval to the actual availability and clinical use of new drugs varies greatly between European Union Member States [3,5,6]. Gann and colleagues observed delays in access to newly registered cancer drugs in some EC of over 4 years [7]. This is worrying as the access to treatment of a disease may affect patient survival, and lack of access conflicts with an individual’s right to health [8]. This right was first laid down in the 1946 Constitution of the World Health Organization and in the 1948 Universal Declaration of Human Rights and ever since is an important cornerstone of many health policies [8].

Access to health care has been defined as “the timely use of services according to needs” [9,10]. Novel drugs are faced with long procedures before patients will have access, not only in the developmental phase, but also in the regulatory processes, after finalization of the pivotal trials. The United States (US) and the European Union (EU) each have their own agencies that provide market authorization for new medicines, respectively the U.S. Food and Drug Administration (FDA) and the European Medicines Agency (EMA). Market authorization is based on the evaluation of safety, efficacy, and quality of the product. Both agencies have special fast track procedures and accelerated approval programs. Moreover, for drugs with high potential patient value, FDA can provide a priority review, that has a maximal review time of 6 months [11]. The accelerated access procedure of EMA should maximally take 150 days, i.e., 5 months [12].

After market authorization, most EC have formal procedures that need to be followed before patients will have access to novel drugs. These procedures commonly include regulatory procedures, price regulations, and some form of health technology assessment to determine whether these drugs will be reimbursed by general means, for instance via a national health services system, or via health insurance schemes [6,13,14,15,16]. Given the increasing pressure on health care budgets, these national procedures are becoming increasingly complex. The procedures and the time they take differ substantially across countries.

Although it is in society’s interest that new drugs, which are proven beneficial to patients, are equally accessible for people in need, it seems to be more and more difficult for EC to strike a balance between benefits and costs of novel cancer drugs [17,18]. As countries cope differently, resulting in variations in patient access, a deeper insight into the problem and its anticipated consequences is necessary.

The aim of this study is to assess variations in national patient access to several newly registered cancer drugs across Europe. Therefore, we compared the dates of submissions to FDA and EMA, the time to first uptake, and speed of uptake of these drugs and explored the impact of observed variations in access in terms of health outcomes.

## 2. Methods

This was a retrospective database study. Data were obtained from the following sources: pharmaceutical sales data was obtained from IQVIA’s MIDAS^®^ database [19]. Sales recorded in MIDAS can originate from both retail or hospital setting. The coverage differs by country and setting. Sales were expressed in standard units (SU)—defined as single tablet or vial—making it impossible to differentiate between dosages. We assumed the usage of varying dosages are similar across included countries. IQVIA’s MIDAS^®^ database did not encompass data on selected drugs for the Netherlands. Dutch data on first uptake were available for all drugs. However, sales data were obtained from manufactures (*n* = 8). We assumed sales data give a good approximation for the usage and access to selected drugs, as it is unlikely that influence of potential stocking of inventories is minimal.

We selected a variety of newly registered cancer drugs. The selection of the drugs was based on diversity in clinical evidence and diversity among indications. We limited our analysis to 12 “end of life medicines” for the indications breast cancer, gastric cancer, prostate cancer, and melanoma. The selected drugs are listed in Table 1. They were first registered between 2011 and 2017 and clinical evidence levels, as determined by the European Society Medical Oncology-Magnitude of Clinical Benefit Scale (ESMO-MCBS) [20], differed. This scale considers outcomes such as (progression-free) survival and drug toxicity. It was hypothesized that the time to patient access may be shorter for drugs with high clinical benefit score (e.g., ESMO-MCBS score 4 or 5) than for drugs with a lower clinical benefit score (e.g., ESMO-MCBS score 2 or 3). Abiraterone, cabazitaxel, vemurafenib, enzalutamide, Palbociclib, and ribociclib had a priority review by FDA. Abiraterone, vemurafenib, and nivolumab underwent a fast track procedure at EMA. 

General and indication-specific cancer data were used for determining the mortality rates per drug indication. Specific cancer mortality data were obtained from Eurostat for the years 2011–2015, mortality for the missing years 2016–2018 was based on extrapolations [21]. Analyses are performed on data from 2010–2018, for 28 European countries (Appendix A).

Subsequently, the time to patient access was determined for each drug. Time to patient access was defined as the sum of: (i) Time from regulatory submission to regulatory approval; (ii) time to first patient access, i.e., time to market (TTM); and (iii) speed of uptake of the drug (Figure 1). 

The “time to market” for 28 European countries was calculated from the date of EMA registration of the drug to the dates of first sales in each country (Figure 1). These dates were defined as dates of first uptake and were obtained from IQVIA’s MIDAS^®^ database [19]. The speed of uptake was calculated by aggregating sales data (in standard units (SU) into the first 24 months of availability in a country and dividing by country-and indication-specific mortality, expressed by the number of cancer (specific) deaths as all drugs were registered for end of life settings. In the case of medicines with multiple indications, data were related to the overall cancer mortality in a country. As in general not all patients are in the appropriate medical condition to receive a new drug, we hypothesized that 80% of the eligible patients should have had access to the drugs.

Thereafter, time to first patient access in the 28 European countries was calculated. For the time of first patient access the date of EMA registration and first uptake in a country were calculated for each drug separately. As sales data are being reported on monthly basis, we assumed that the first uptake date would always be on the 1st of every month. Thereafter, these number of days were averaged for all 12 drugs.

Additionally, the speed of uptake in a country has been calculated by using the following formula:(1)$$Speed\;of\;uptake\;drug\;in\;country\;=\overset{n=12}{\underset{n=1}{\sum }}\left(\frac{sales\;volume\;drug\;after\;1\;and\;2\;years}{mortality\;of\;drug\;indication\;in\;these\;years}\right)$$
*n* = type of drug, 12 drugs included in the analysis.

The sales volumes were calculated by summing up the sales volumes after exactly 1 and 2 years after the date of first uptake per drug per country. The outcomes were divided by the mortality that corresponded to the drug indication and the year. Thereof the average rank of all studied drugs per country has been derived.

To illustrate the impact of delay in patient access in European countries, we selected ipilimumab and abiraterone, as these drugs have a high clinical value (ESMO 4) and the trial results have shown an impact on the overall survival, namely an increase by 3.7 months and 3.9 months, respectively [22,23]. We calculated the loss in life years (LYs) due to a delayed access in their first year after market approval as for both drug indications new comparators were introduced later in time. We also estimated the loss in LYs due to a later introduction in Europe as compared to the US. For the number of patients in need for abiraterone and ipilimumab we used the dosing and the median number of cycles from the clinical trials [23,24]. The latter was related to the time to disease progression.

Further, the relation between FDA or EMA and between the ESMO-MCBS on the time to market and the speed of uptake has been studied by means of regression analyses (ANOVA). The ESMO-MCBS score was based on the results of the first publication. All statistical analyses were performed in SPSS Statistics version 25 for Windows (SPSS Inc. Chicago, IL, USA).

## 3. Results

Table 1 and Appendix A show the dates of the submission to, and approval by the EMA and FDA. The dates of submission to EMA and FDA were almost comparable, with the exception of palbociclib (395 days later in Europe). All drugs were first approved in the US. On average, the time to first registration was 181 days in the US (range 78–303 days) vs. 378 days in Europe (range 262–483 days), implying a difference in duration of the procedures of 197 days. Marketing approval for the cancer drugs came on average 242 days later in Europe than in the US. For the three drugs assessed in the accelerated trajectory of EMA, the average assessment period was 280 days. For drugs in the standard trajectory, this period was 410 days. The 6 drugs undergoing priority review by FDA, took an average time to market approval of 139 days, compared to 223 days for the drugs in the regular trajectory.

In Figure 2, the EMA trajectory is presented per studied drug. The actual EMA assessment time averaged 204 days and the time the applicants needed to answer queries averaged 86 days. The time between submission of the dossier and the start of the regulatory assessment procedure averaged 27 days, while the time between a positive opinion and approval averaged 61 days.

Further, there was no relation found in time between registration by FDA or EMA, and clinical value of the drugs as defined by clinical outcomes (OS, PFS, or TTP), or ESMO-MCBS score. For example, ipilimumab resulted in a gain of 3.7 months in OS and had an ESMO-MCBS score 4 and it took EMA 433 days to approve (FDA: 278 days). In contrast, for cabazitaxel, with 2.4 months increase in time to progression and ESMO-MCBS score 2 market authorization was given 331 days after submission of the EMA dossier (FDA priority review: 78 days).

Figure 3 and Appendix B present the average time from EMA registration to first uptake of the studied drugs across Europe. 2–8 Years after marketing approval, several countries still either had a very low uptake of drugs, or no uptake at all. Palbociclib had the fastest time to market from EMA registration until first uptake in the EC (average: 165 days), followed by nivolumab (average: 210 days), but 2 years after European market approval, these drugs were still not prescribed to patients in four and five countries, respectively. Note that, despite the relatively fast uptake of palbociclib, the time between US and EU market access was almost two years. For nivolumab this period was shorter, namely 179 days.

The average TTM in Europe amounted to 398 days (range 17–1187 days). In general, patients in Germany, the UK, and Austria had the most rapid potential access, with averages of 17, 22, and 31 days, respectively. Greece and many Eastern European countries were below the European average.

Figure 4 shows the speed of uptake of drugs 2 years after approval in a country. Belgium, Switzerland, France, and Austria had the highest uptake after two years. The UK and Sweden had relatively slow uptakes after 2 years. 

Concerning the time to first uptake in Eastern EC, Poland was fastest, followed by Slovakia and Slovenia. First patient access to the drugs in these countries was faster than, for instance, Spain, Ireland and Italy. Bulgaria, Romania, Croatia, and Latvia ranked low in time to first access, but both Bulgaria and Czech Republic thereafter had a rapid uptake. 

A delay in patient access to new drugs may result in diminished patient benefits. We calculated that in Europe approximately 14,994 patients were eligible for treatment with ipilimumab in the first year after EMA approval (see Table 2 and Appendix C). Taking into account the sales per country in that first year approximately 11,184 melanoma patients were not treated with ipilimumab. Assuming an average gain in OS of 3.7 months (derived from Table 1), this may have resulted in a loss of 3448 life years. Applying the same calculation to prostate cancer patients eligible for abiraterone resulted in 55,853 non-treated patients, which would indicate a loss of 18,152 life years across Europe for abiraterone non-use. The delay in the EMA time to registration compared to the FDA time led to an extra estimated loss of 8639 life years for both drugs.

## 4. Discussion

The results of our study show that, although the dates of submission to EMA and FDA did not differ for most drugs, on average newly registered cancer drugs entered the European market eight months later than the USA market. Moreover, time to patient access to the 12 newly registered cancer drugs included in the analyses differed strongly across Europe. Our analysis is the first showing the potential impact of a delay in access for patients. In the first year after EMA market authorization of ipilimumab and abiraterone almost 67,000 patients were unable to benefit from these drugs, resulting in an estimated loss of 21,600 life years. The longer EMA time to registration, as compared to the FDA time to registration, led to an extra estimated loss of 8693 life years.

Wilking and Jonsson previously studied patients’ access to treatments in the five most common tumor types for the period 1999–2004 [5]. In that period Austria, Spain, and Switzerland were fastest in realizing patient access. As in our study the UK was quite slow in adoption of the cancer treatments. Another study compared the uptake and market share for direct acting antivirals in six European countries [25]. In Germany and France patients had early access and these countries were fast adopters of these drugs. Spain and Italy were late in first uptake, but they were fast adopters after first uptake. In the UK, patients had fast access, but the uptake was slow.

As all European countries cope differently with newly registered drugs, resulting in variation in patient access, a deeper understanding of the facilitators, barriers, and key actors involved in this process is necessary. According to Frost and Reich, access to an innovation depends of several factors, such as availability, affordability, and adoption of the intervention [26]. The availability of a newly registered drug in a country will be influenced by factors like time of market authorization, the duration of the reimbursement procedure and health technology assessment, the used pricing system (e.g., external reference pricing (ERP)) and the value of the drug. Affordability means that the drug is not too expensive. This is mainly influenced by the price, the gross domestic product (GDP) of a country, the health care expenditure of a country, the pharmaceutical spending of country, and the financing (co-payments or limits on number of patients treated). Adoption depends on the acceptance and amount of unmet need of the intervention as perceived by several actors, such as global organizations (FDA or EMA), governments, doctors, and individual patients. Further study of the facilitators, barriers, and key actors involved in the access of new drugs are highly recommended.

Recently, several methods have been developed in Europe and the US to deal with the assessment of the value and pricing of newly registered drugs, and their affordability in the health systems. Examples are the American Society for Clinical Oncology (ASCO) Value in Cancer Care Framework and the ESMO-MCBS [18,27]. These methods focus on the clinical benefit of the drugs and (partly) on value-based pricing, addressing cost or cost-effectiveness of the new drug. In this study we have used the ESMO-MCBS to assess the clinical value of the studied drugs, but other instruments could be used as well. We expected that higher values of the ESMO-MCBS would result in a faster access. However, in our study a higher value, i.e., ESMO-MCBS 4–5, did not lead to a faster access of patients to these drugs.

Our study has a number of limitations. First, this study was based on data from several retrospective data sources. Each data source has several strong and weak points. IQVIA’s MIDAS^®^ database includes worldwide standardized sales data allowing unique cross-country comparisons over time. However, in some countries not all distribution channels (e.g., hospital/retail) are captured and the database does not include direct sales to clinics and private offices in all countries. Moreover, data coverage differs per country, which despite regular quality and validity checks, potentially impacts accuracy of data extrapolations.

Second, there may be differences in the quality of the registrations of cancer mortality in the EC. Some countries may have more reliable data than other countries. However, the methods to calculate the mortality rates are standardized. 

Third, some drugs had registrations for the same indication or for a specific sub-indication (e.g., melanoma for patients with PDL-1 expression) and could be used as substitutes. Further, some drugs are used for multiple indications (e.g., nivolumab: lung cancer, melanoma). In case of multiple indications, we used overall cancer mortality rates of the countries to compare the uptake. As a result, we could not calculate the exact loss in life years as a result of the delay in access of patients to these treatments. Loss in palliative effect of the drugs (i.e., lost potential effects on quality of life rather than survival) is something we could also not assess.

Fourth, data about uptake of drugs should ideally be collected by using registry data, capturing data on patient and disease characteristics, and real-world use of the drugs (dosing and number of days/cycles). In the absence of such data in Europe, we used data from IQVIA MIDAS, Eurostat, and clinical trials [19,21,22,23]. Data on speed of uptake were based on sales data and on country specific cancer mortality rates as the drugs were end of life products. We estimated that 80% of the patients in real life were eligible for the drugs, as some patients would be too unfit and/or would have too many co-morbidities to enable treatment. For the number of patients in need for abiraterone and ipilimumab we used data from the clinical trials. It is possible that in clinical practice patients may receive fewer cycles, implying that more patients may have received these drugs. If so, this has resulted in a slight overestimation of loss of life years.

Fifth, the inclusion of drugs was based on a pragmatic approach. A different selection of drugs may have resulted in different time to access estimated. Moreover, this study was focused on patient access to oncology drugs. Time to access and uptake may be different in other disease areas. 

Finally, we selected two drugs to give an illustration of life years lost in Europe due to delays in patient access. The estimation of life years lost is based on a high-level calculation. It would be worthwhile to conduct a study including more drugs and more elaborate calculations.

Time to patient access in Europe is influenced by the complexity of national reimbursement processes. Most pharmaceutical companies first launch their product in Germany as it is the largest European market and reimbursement is automatic once EMA has approved drugs. A year of free pricing is allowed while price and reimbursement negotiations are ongoing [28]. Countries in which the reimbursement is dependent on the outcome of cost-effectiveness assessments (e.g., UK and The Netherlands) or in which lengthy negotiations with national and regional decision-makers have to take place (e.g., Spain and Italy) take a longer time to first access and have more limited uptake after two years. We assumed market access to be similar in all countries because of the centralized EMA procedure, however Norway and Switzerland have their own agencies, resulting in a 75-and 66-day delay, respectively [29,30]. Therefore, time to patient access in Norway and Switzerland has been slightly overestimated.

Several aspects can help shortening the time to patient access and increase uptake.

Specific early access programs can help facilitate early launches as exemplified in France, Sweden, and Italy [27]. Since the current processes of early access programs are generally complex, governments may be able to better facilitate these programs, for instance by allowing pharmaceutical companies to provide the medicines for free during the process of price negotiations and to reimburse the drugs according to the negotiated price once the negotiations have ended. The FDA assessment was on average substantially faster than the EMA assessment. This was during the whole study period (2010–2018). Therefore, improvements in the EMA procedure seem possible [31]. For instance, shortening the time from EMA submission to procedure start and the time from positive opinion to approval may accelerate the process by almost 3 months.

The coming decade, the number of patients with cancer is estimated to increase by 68% [32]. As stated before, patients have a right to health, i.e., the highest attainable standard of health as a fundamental right of every human being [8]. This makes it a legal obligation of countries to ensure timely access to acceptable and affordable health care of appropriate quality [32]. Fortunately, this issue will be addressed in the Pharmaceutical Strategy for Europe commissioned by the European Commission [33]. As many novel cancer drugs have entered the market and many others are upcoming, it is of utmost importance that all patients in need get access to the drugs with high clinical value as soon as possible.

## 5. Conclusions

This study shows that it takes a long time for European patients to get access to newly registered cancer drugs. Further, there is great inter-country variation of access to new cancer drugs. The delay in access may result in a potential loss of many life years. The health outcomes of European patients can substantially be improved by enabling faster and more general use of available new medicines.

## Figures and Tables

**Figure 1 cancers-12-02313-f001:**
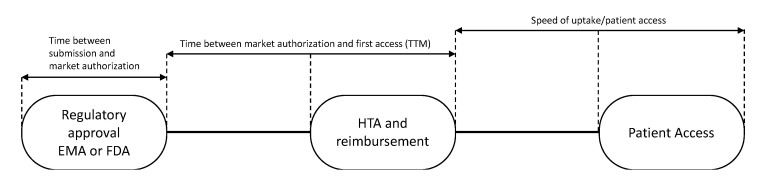
Patient newly registered drug access pathway. EMA: European Medicines Agency; FDA: USA Food and Drug Association; HTA: health technology assessment; TTM: time to market.

**Figure 2 cancers-12-02313-f002:**
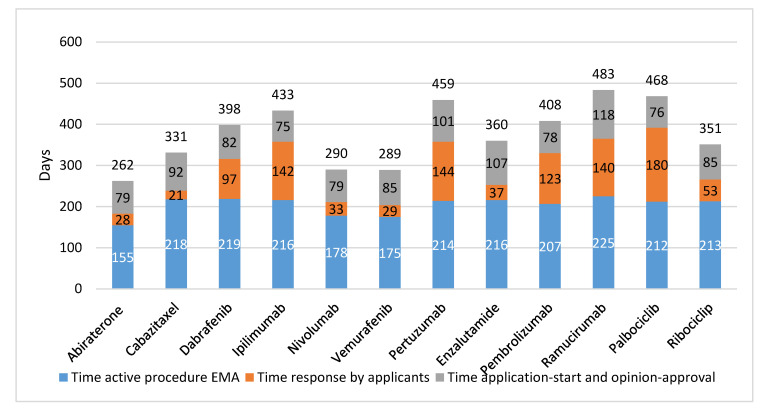
EMA trajectory of 12 newly registered oncological drugs (in days). EMA: European Medicines Agency.

**Figure 3 cancers-12-02313-f003:**
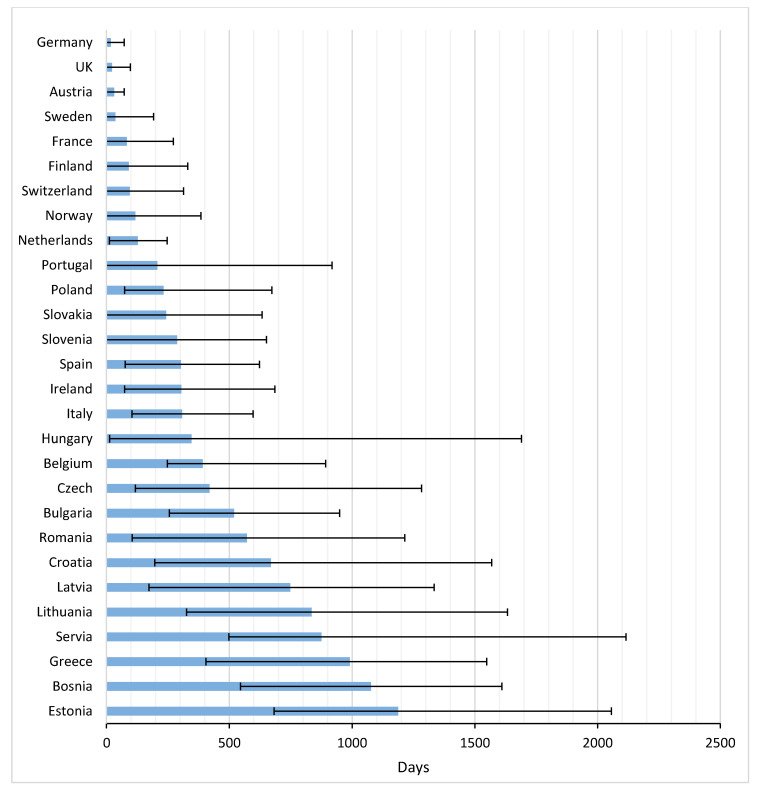
Time to first uptake for 12 newly registered oncological drugs across Europe (in days).

**Figure 4 cancers-12-02313-f004:**
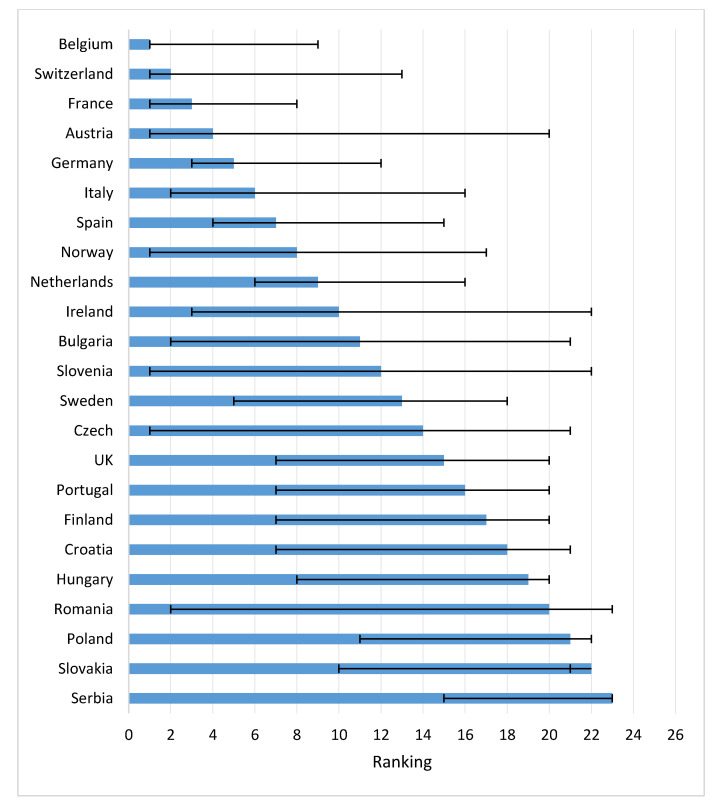
Speed of drug uptake for 12 newly registered oncological drugs in first two years across Europe (average rank, range) (Note: Too little access data for ranking: Lithuania, Greece, Bosnia, Estonia).

**Table 1 cancers-12-02313-t001:** Newly registered oncological drugs, first indications, clinical values, and duration of EMA and FDA procedures.

Drug	First Indication	Gain PFS, OS, TTP (Median, Months)	ESMO-MCBS *	Date of EMA Submission	Date of EMA Approval	Accelerated Assessment (EMA)	Total Time EMA (in Days)	Date of FDA Submission	Date of FDA Approval	Priority Review (FDA)	Total Time FDA (in Days)	Time between EMA and FDA Approval (in Days)
Abiraterone	Prostate cancer	3.9 months OS	4	17 December 2010	5 September 2011	16 December 2010	262	20 December 2010	28 April 2011	Yes	129	130
Cabazitaxel	Prostate cancer	2.4 months TTP	2	20 April 2010	17 March 2011	n.a.	331	31 March 2010	17 June 2010	Yes	78	273
Dabrafenib	Melanoma	2.4 months PFS	4	24 July 2012	26 August 2013	n.a.	398	30 July 2012	29 May 2013	No	303	89
Ipilimumab	Melanoma	3.7 months OS	4	05 May 2010	12 July 2011	n.a.	433	10 June 2010	15 March 2011	No	278	119
Nivolumab	Melanoma	4.0 months PFS	4	02 September 2014	19 June 2015	24 July 2014	290	30 July 2014	22 December 2014	No	145	179
Vemurafenib	Melanoma	3.7 months PFS	4	04 May 2011	17 February 2012	14 April 2011	289	28 April 2011	17 August 2011	Yes	111	184
Pertuzumab	Breast cancer	6.1 months PFS	4	01 December 2011	4 March 2013	n.a.	459	06 December 2011	08 June 2012	No	185	269
Enzalutamide	Prostate cancer	4.8 months OS	4	26 June 2012	21 June 2013	n.a.	360	22 May 2012	31 August 2012	Yes	101	294
Pembrolizumab	Melanoma	1.3 months PFS	3	04 June 2014	17 July 2015	n.a.	408	27 February 2014	03 September 2014	No	188	317
Ramucirumab	Gastric cancer	2.2 months OS	2	23 August 2013	19 December 2014	n.a.	483	23 August 2013	21 April 2014	No	241	242
Palbociclib	Breast cancer	10.3 months PFS	3	30 July 2015	9 November 2016	n.a.	468	30 June 2014	03 February 2015	Yes	218	645
Ribociclib	Breast cancer	PFS not reached	3	05 September 2016	22 August 2017	n.a.	351	29 August 2016	13 March 2017	Yes	196	162
Average time (in days)							378				181	242
Average time accelerated assessment/priority review (in days)							280				139	n.a.
Average time in case no accelerated assessment/no priority review (in days)							410				223	n.a.

* Ref. [22] PFS: progression-free survival; OS: overall survival; TTP: time to progression, ESMO-MCBS: European Society Medical Oncology-Magnitude of Clinical Benefit Scale; EMA: European Medicines Agency; FDA: USA Food and Drug Association; n.a.: not applicable.

**Table 2 cancers-12-02313-t002:** Potential life years lost due to delay in access in abiraterone and ipilimumab across Europe.

Country	Abiraterone			Ipilimumab		
	Difference	Delay in Access	Total	Delay in Access	Difference	Total
	in Track	after EMA	Life	after EMA	in Track	Life
	FDA-EMA	Registration	Years Lost	Registration	FDA-EMA	Years Lost
Austria	115	204	318	31	50	81
Belgium	140	376	516	26	69	95
Bulgaria	89	249	338	13	40	53
Croatia	72	203	275	15	46	62
Czech Republic	157	440	597	38	114	152
Estonia	25	70	95	4	14	18
Finland	85	234	319	18	50	68
France	854	1803	2657	150	185	336
Germany	1126	2466	3592	219	394	613
Hungary	119	334	453	33	100	133
Ireland	84	234	318	17	44	61
Italy	602	1691	2293	143	433	576
Latvia	37	104	141	6	19	25
Lithuania	55	155	211	8	26	34
Netherlands	292	733	1025	72	194	266
Norway	117	273	390	31	94	125
Poland	507	1416	1923	127	385	512
Portugal	164	456	621	21	63	84
Romania	225	632	857	36	45	81
Serbia	102	287	389	22	68	90
Slovakia	92	256	349	19	58	77
Slovenia	41	113	155	11	32	42
Spain	580	1545	2126	79	240	318
Sweden	235	583	818	46	132	178
Switzerland	136	305	440	32	57	89
United Kingdom	1170	2988	4159	200	495	695
Total life years lost	7221	18,152	25,373	1418	3448	4867

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
