# Peer review of "Unequal Access to Newly Registered Cancer Drugs Leads to Potential Loss of Life-Years in Europe"

_cancers, 2020, doi:10.3390/cancers12082313_

Round 1

Reviewer 1 Report

This manuscript describes a concerning issue related to inequities on innovation access between Europe and US. The design of this study is also original, and the conclusions were not surprising.

Title:  Should be informative and concise about the study but not being the conclusion heading. Please rephrase.

INTRODUCTION

-Please revise the description of the objective of this study, and redefine according to the results and conclusions. Must to be clear and concise, and must be the question and the answers are results and conclusions.

METHODS

-Justify more the election of the 12 drugs. Clarify which indications were analysed in terms of impact on PFS and OS.

-Statistical section is lacking. Despite not finding relation between registration by FDA or EMA, and clinical value of the drugs, it is mandatory to explain how the authors carried out the statistical analysis.

DISCUSSIOn

-The major part of this section corresponds to limitations. Please discus your results with studies of such importance like ESMO one for instance and others regarding different access among European countries.

Author Response

Thank you for your comments and positive feedback. We have answered your questions consecutively.

“Title:  Should be informative and concise about the study but not being the conclusion heading. Please rephrase.”

Response: We have discussed this comment internally, but still prefer our original title. In our opinion, this title is more challenging, which will result in more interested readers. We have looked up other titles in CANCERS and noticed that both options are used, namely informative titles about the study and titles which already shows the main conclusion. If the reviewer still want us to the change our title, we will of course do it. In that case, we suggest to adapt the title in “Access to Newly registered Cancer Drugs in Europe: an analysis of variation and impact on health outcomes of patients”.

 INTRODUCTION

“-Please revise the description of the objective of this study, and redefine according to the results and conclusions. Must to be clear and concise, and must be the question and the answers are results and conclusions.”

Response: We have adapted the objectives in “The aim of this study is to assess variations in national patient access to several newly registered cancer drugs across Europe. Therefore, we compared the dates of submissions to FDA and EMA, the time to first uptake and speed of uptake of these drugs and explored the impact of observed variations in access in terms of health outcomes.”

 METHODS

“-Justify more the election of the 12 drugs. Clarify which indications were analysed in terms of impact on PFS and OS.”

Response: We have added and adapted the text in: “We selected a variety of newly registered cancer drugs. The selection of the drugs was based on diversity in clinical evidence and diversity among indications. We limited our analysis to 12 ‘end of life medicines’ for the indications breast cancer, gastric cancer, prostate cancer and melanoma.” We did not analyse the impact on PFS. We only choose to 2 drugs, which have an impact on OS.

See Page 4,line 30: To illustrate the impact of delay in patient access in European countries, we selected ipilimumab and abiraterone, as these drugs have a high clinical value (ESMO 4) and the trial results have shown an impact on overall survival, namely an increase by 3.7 months and 3.9 months, respectively.

-Statistical section is lacking. Despite not finding relation between registration by FDA or EMA, and clinical value of the drugs, it is mandatory to explain how the authors carried out the statistical analysis.

Response:  See adaptations in methods. “Further, the relation between FDA or EMA and between the ESMO-MCBS on the time to market and the speed of uptake has been studied by means of regression analyses (ANOVA). The ESMO-MCBS score was based on the results of the first publication. All statistical analyses were performed in SPSS Statistics version 25 for Windows (SPSS Inc. Chicago, IL).

Reviewer 2 Report

The article submitted by Groot et al entitled "Unequal Access to Newly registered Cancer Drugs leads to potential loss of life-years in Europe" presents an analysis of the time required for patient access for new cancer drugs in US and in Europe. Different parameters are evaluated, including time to patient access, speed of drug uptake and potential loss of life years due to delay in access. Analysis are based on 12 cancer drugs in the period 2011-2018 and involve 28 countries. The articles is very interesting and of great potential impact.

While the emphasis is given to the differences between Europe and US, the study presents a number of important aspects that should be brought into the attention of law makers and health agencies in the different countries.

The results show the existence of significant average differences between US and Europe for all the 12 cancer drugs considered. However, what is strikingly large are the reported differences among the various european countries.

While EMA approval takes longer than FDA approval for all drugs considered, it is a common stage for all the different countries in Europe. It does not justify the astonishing large differences reported between patients in Germany, Austria, UK or Sweden, for example, with those in countries like Portugal, Spain or Italy, or even more dramatically Greece and some eastern europen countries. These points could be explored to a greater extention.

In particular, it could be analyzed the existence of a relation between this delay and other social-economical aspects, including national GDP, taxes, number of doctors per 100,000 inhabitants, educacional level, % of PIB invested in health, etc.

Other points that shoud be included in the discussion:

  • The difference between required time for FDA and EMA approval are quite different. Is this difference in time decreasing in more recent years? Or increasing? How would it compare with data from the early 2000s? Could this effect be evaluated
  • Are these trends illustrative of the entire process of drug acess to patients? Or specific to cancer drugs
  • -Even withouth detailed data, what do the authors anticipate would be the scenario in other countries? Japan? Australia, New Zealand ? Asia?

Minor Issues:

- Figure 2 could be improved for clarity

Author Response

Thank you for your comments and positive feedback. We have answered your questions consecutively.

“While EMA approval takes longer than FDA approval for all drugs considered, it is a common stage for all the different countries in Europe. It does not justify the astonishing large differences reported between patients in Germany, Austria, UK or Sweden, for example, with those in countries like Portugal, Spain or Italy, or even more dramatically Greece and some eastern European countries. These points could be explored to a greater extension.”

Response: We have added an additional paragraph in the discussion. The next step will be to perform logistic regressions to check whether there is a relation between this delay and other social-economical aspects. An issue will be whether these aspects could be influenced or not in order to improve access to newly registered drugs.

“Other points that should be included in the discussion: The difference between required time for FDA and EMA approval are quite different. Is this difference in time decreasing in more recent years? Or increasing? How would it compare with data from the early 2000s? Could this effect be evaluated”

Response: There is no difference in time in- or decreasing in more recent years. However, we did not study data from earlier years, i.e. before 2011. We have added a sentence to deal with this point (page13, lines 197-198)).

“Are these trends illustrative of the entire process of drug access to patients? Or specific to cancer drugs.”

Response: Our focus was on cancer drugs, but there is also literature about other indications/diseases.

See page 8, line 112.

“-Even without detailed data, what do the authors anticipate would be the scenario in other countries? Japan? Australia, New Zealand ? Asia?”

 Response: We anticipate that the uptake in other countries will be equal different or worse (especially in less developed countries).

Minor Issues:

“- Figure 2 could be improved for clarity”

Response: We have adapted this figure and adapted figure 3 and 4 as well (now in color).